# Diversity of a Lactic Acid Bacterial Community during Fermentation of Gajami-Sikhae, a Traditional Korean Fermented Fish, as Determined by Matrix-Assisted Laser Desorption/Ionization Time-of-Flight Mass Spectrometry

**DOI:** 10.3390/foods11070909

**Published:** 2022-03-22

**Authors:** Eiseul Kim, Ji-Eun Won, Seung-Min Yang, Hyun-Jae Kim, Hae-Yeong Kim

**Affiliations:** Department of Food Science and Biotechnology, Institute of Life Sciences & Resources, Kyung Hee University, Yongin 17104, Korea; eskim89@khu.ac.kr (E.K.); wldms7183@naver.com (J.-E.W.); ysm9284@gmail.com (S.-M.Y.); tkdzmtm2@naver.com (H.-J.K.)

**Keywords:** gajami-sikhae, MALDI-TOF MS, microbial community, culture-dependent method, fermentation, identification, fermentation temperature

## Abstract

Gajami-sikhae is a traditional Korean fermented fish food made by naturally fermenting flatfish (*Glyptocephalus stelleri*) with other ingredients. This study was the first to investigate the diversity and dynamics of lactic acid bacteria in gajami-sikhae fermented at different temperatures using matrix-assisted laser desorption/ionization time-of-flight mass spectrometry (MALDI-TOF MS). A total of 4824 isolates were isolated from the fermented gajami-sikhae. These findings indicated that *Latilactobacillus*, *Lactiplantibacillus*, *Levilactobacillus*, *Weissella*, and *Leuconostoc* were the dominant genera during fermentation, while the dominant species were *Latilactobacillus* *sakei*, *Lactiplantibacillus* *plantarum*, *Levilactobacillus* *brevis*, *Weissella* *koreensis*, and *Leuconostoc* *mesenteroides*. At all temperatures, *L. sakei* was dominant at the early stage of gajami-sikhae fermentation, and it maintained dominance until the later stage of fermentation at low temperatures (5 °C and 10 °C). However, *L*. *plantarum* and *L*. *brevis* replaced it at higher temperatures (15 °C and 20 °C). The relative abundance of *L. plantarum* and *L. brevis* reached 100% at the later fermentation stage at 20 °C. These results suggest that the optimal fermentation temperatures for gajami-sikhae are low rather than high temperatures. This study could allow for the selection of an adjunct culture to control gajami-sikhae fermentation.

## 1. Introduction

Sikhae is a traditional fermented food in Korea, commonly served as a side dish. Gajami-sikhae, which uses flatfish belonging to the species *Glyptocephalus stelleri* as the main ingredient, is made by mixing salted flatfish with ingredients such as cooked grains, salt, red pepper powder, white radish, garlic, ginger, and green onion [1]. In recent years, gajami-sikhae has been well accepted by Korean consumers due to its unique flavor and potential health benefits, such as anticancer and antioxidant effects, attributed to the fermentation of various microorganisms [2,3].

The spontaneous fermentation of foods is mainly affected by the microorganisms present in the food at various stages of the fermentation process [4]. Lactic acid bacteria (LAB) are the most prominent microorganisms responsible for fermenting vegetables, meat, dairy products, and fish [5]. Some studies showed that the main taxa of fermented fish are related to *Latilactobacillus* and *Weissella*, including *Latilactobacillus sakei* and *Weissella koreensis* [6,7]. The quality of fermented foods correlates to the various microorganisms that occur naturally during the fermentation process; they produce bacteriocins, organic acids, and flavor compounds responsible for the flavor formation of fermented foods [8]. Since the growth of LAB in fermented foods is affected by fermentation conditions, such as the fermentation temperature and period, it is necessary to investigate the change in the microbial community under various fermentation conditions to improve the quality of gajami-sikhae made for consumption.

Investigations of the entire microbial community present in many foods became possible with the advent of next-generation sequencing [4]. This technology has been successfully applied to study microbial communities in a fermented food matrix. Reportedly, it provides deeper, more precise information on the microbial community than polymerase chain reaction denaturing gradient gel electrophoresis (PCR-DGGE) [4,9]. Additionally, there are approaches to investigate viable microbial communities using metagenetics (e.g., after total RNA extraction and reverse PCR or through the use ethidium monoazide (EMA) treatment PCR to only amplify DNA from viable cells) [10]. However, the species identification obtained using the metagenomic approach might be limited.

Culture-based approaches are still widely used to analyze microbial communities. The development of new tools, such as matrix-assisted laser desorption/ionization time-of-flight mass spectrometry (MALDI-TOF MS), has allowed a reduced time to detection as compared to conventional culture-based methods [11]. Currently, MALDI-TOF MS is an alternative to the sequencing method for identifying microorganisms [12]. This high-throughput technique compares mass spectral patterns, including ribosomal protein obtained from microbial cells, with a reference spectral database [13]. This technique has emerged as a new method for the relatively rapid, simple, and effective identification of microorganisms based on its reliance on microbial fingerprints [12]. Moreover, MALDI-TOF MS is superior to the 16S rRNA gene for taxonomic resolution at the species or subspecies level for some closely related species such as *Lacticaseibacillus casei*/*L*. *paracasei* and *Lactobacillus acidophilus* group species [14,15,16,17]. However, MALDI-TOF MS relies on a spectral database, so only species present in the database can be identified.

Many studies addressing its use in experimental approaches related to pathogenic bacteria have been published [12,13]. Some studies have applied MALDI-TOF MS technology to observe changes in the culturable microbial community in fermented foods, but there has been no study examining the microbial community of gajami-sikhae [18]. Moreover, although the microbial community of gajami-sikhae has been investigated by pyrosequencing [19], the effect of temperatures on this fermentation process has not yet been investigated.

In this study, we aimed to analyze the microbial community during the fermentation of gajami-sikhae using MALDI-TOF MS. The results provide a deeper understanding of the correlation between fermentation conditions and the microbial community. In addition, they will lay a foundation for standard gajami-sikhae manufacturing systems and quality improvement.

## 2. Materials and Methods

### 2.1. Sample Preparation

A gajami-sikhae sample was purchased from traditional manufacturers in Korea in December 2020. A sample prepared using the traditional method, in which the salted flatfish (*Glyptocephalus stelleri*) was mixed with radish, red pepper powder, boiled millet (*Setaria italica*), chopped garlic, and NaCl, was purchased [19]. This mixture was fermented at 5 °C, 10 °C, 15 °C, or 20 °C for 60 days in a poly cyclohexane-1,4-dimethylene terephthalate plastic bowl (20 × 11 cm). Samples were collected at from 8 to 11 points at each fermented temperature. Thirty-nine samples in total were obtained. Appendix A provides detailed information about the obtained samples.

### 2.2. pH and Acidity Measurements

The gajami-sikhae sample was ground for 2 min using a blender and filtered through gauze to remove large particles and measure the pH and acidity. The pH value of the filtered gajami-sikhae samples (50 mL) was measured in triplicate using a pH meter (Thermo Fisher Scientific, Waltham, MA, USA). Also, 10 mL of the filtered sample was titrated with 0.1 N NaOH to a final pH of 8.2 to measure the acidity of the gajami-sikhae [20,21]. The acidity was calculated by substituting the measured volume of 0.1 N NaOH into the percentage (%, *v*/*v*) of lactic acid produced.

### 2.3. Cultivable Microbial Community

#### 2.3.1. Isolation of LAB

For the isolation of LAB, 25 g of gajami-sikhae sample and 225 mL of sterilized phosphate-buffered saline (PBS) were placed in a sterile stomacher bag (Seward Limited, London, UK). The mixture was homogenized for 2 min at 230 rpm using a peristaltic homogenizer (Circulator stomacher 400; Seward Limited). Subsequently, serial dilutions of the homogenate were prepared, followed by isolation of LAB on MRS (Difco) agar incubated at 20 °C and 30 °C for 72 h. After incubation, the colony-forming units (CFU) were counted. All colonies from countable plates with lactic acid bacterial growth of between 30 and 300 CFU/plate on MRS agar were selected. The harvested colonies were subsequently subcultured on MRS agar and incubated under similar conditions to those described above.

#### 2.3.2. Analysis of the Microbial Community by MALDI-TOF MS

A single colony was smeared onto a polished steel MALDI target plate (Bruker Daltonics, Bremen, Germany). The spot was covered with 1 µL of 70% formic acid and dried at room temperature. Subsequently, 1 µL HCCA matrix solution containing 10 mg/mL α-cyano-4-hydroxycinnamic acid (CHCA) (Bruker Daltonics) in acetonitrile, water, and trifluoracetic acid (50:47.5:2.5 (*v*/*v*/*v*)) was added to the spot and dried again. The polished steel MALDI target plate was introduced into the Microflex LT bench-top MALDI-TOF mass spectrometer (Bruker Daltonics). The mass spectra of isolates were identified by comparing the mass spectra to those in the Bruker MSP database version 4.0, containing 5627 reference spectra, using the Bruker software. The Bruker MSP database consists of 98 species and 236 spectra of LAB (*Lactobacillus*-related species). The identification score was interpreted according to the manufacturer’s criteria. Thus, a score between 2.0 and 3.0 indicated highly probable species identification, between 1.7 and 1.999 indicated probable genus identification, and lower than 1.7 indicated unreliable identification. For lactic acid bacterial identification, the following reference strains obtained from the Korean Agricultural Culture Collection (KACC, Jeonju, Korea) and the Korean Collection for Type Cultures (KCTC, Daejeon, Korea) were used: *Lactiplantibacillus plantarum* KACC 11451, *L*. *sakei* KCTC 3603, *Latilactobacillus curvatus* KACC 12415, *Levilactobacillus brevis* KCTC 3498, *Leuconostoc mesenteroides* KCTC 3100, *Leuconostoc inhae* KACC 12281, *Leuconostoc gelidum* KACC 12256, *Weissella cibaria* KCTC 3746, *Weissella confusa* KCTC 3499, and *W*. *koreensis* KACC 11853.

### 2.4. Statistical Analysis

The pH and acidity values are expressed as the means ± standard deviations. The statistical analysis for pH and acidity values was performed using R v.4.1.0. Significant differences (*p* < 0.05) between the sample means were determined by Duncan’s multiple range test. In addition, the relationships between the major species in fermented gajami-sikhae and the pH, acidity, fermentation temperature, and fermentation period were determined by calculating the Pearson correlation implemented in R. Benjamini–Hochberg correction was used to correct the *p* values [22].

## 3. Results and Discussion

### 3.1. Physicochemical Properties

The pH and acidity affect fermented foods’ anaerobic fermentation efficiency and should be monitored during food fermentation [23]. During fermentation, the gajami-sikhae tended to decrease in pH and increase in acidity as the fermentation progressed. Figure 1A shows the change in pH during the fermentation of gajami-sikhae. The pH value was 6.25 ± 0.01 immediately after gajami-sikhae production and further dropped during fermentation (Appendix A). During the first 15 days, the pH values of samples fermented at 5 °C and 10 °C rapidly decreased to 4.54 ± 0.01 and 4.44 ± 0.01, respectively. However, the pH value remained stable from then until the end of fermentation. In samples fermented at 15 °C and 20 °C, the pH values rapidly decreased to 4.81 ± 0.01 and 4.50 ± 0.01, respectively, in the first two days and gradually decreased thereafter. The pH value of gajami-sikhae at 20 °C was the lowest after fermentation (Figure 1A).

The acidity was 0.33 ± 0.03% at the beginning of fermentation and then increased during the fermentation process at all temperatures (Figure 1B). The acidity of gajami-sikhae fermented at 5 °C and 10 °C rapidly increased and reached about 1.18 ± 0.02% and 1.32 ± 0.01%, respectively, after 20 days. From this point, it remained stable. After 50 days, the acidity value of gajami-sikhae fermented at 15 °C gradually increased to 1.86 ± 0.03%. Also, the acidity of gajami-sikhae fermented at 20 °C continued to increase, showing 2.77 ± 0.02% acidity at 50 days of fermentation. During fermentation, gajami-sikhae fermented at relatively low temperatures (5 °C and 10 °C) showed low acidity, whereas samples fermented at high temperatures (15 °C and 20 °C) showed high acidity. Compared with the control, a lower pH value occurred in the sample fermented at 20 °C, indicating that the fermentation temperature affected the acidity of the gajami-sikhae. Fermented food quality is usually unacceptable when the acidity is about 1.6–2.0% [9]. In this study, the acidities at 15 °C and 20 °C reached unacceptable levels after 10 and 15 days, respectively. However, samples fermented at 5 °C and 10 °C did not reach unacceptable acidity (1.6–2.0%) until the end of the fermentation period.

The LAB population of the gajami-sikhae was estimated by plate counting on MRS agar. Our findings indicated that the number of viable cells rapidly increased and then slightly decreased throughout the fermentation. The initial lactic acid bacterial count in gajami-sikhae was 4.0 log CFU/g. Also, at 5 °C and 10 °C, LAB counts reached the maximum in 10 days, with an average of 8.2 log CFU/g and 8.0 log CFU/g, respectively. Then, the counts slightly decreased until the end of fermentation (Figure 2A,B). Finally, samples fermented at 15 °C and 20 °C reached the maximum cell counts with an average of 8.4 CFU/g and 7.9 CFU/g after three and two days of fermentation, respectively.

### 3.2. Identification of Isolates Using MALDI-TOF MS

The traditional culture method is often intensive and time-consuming. However, alternative molecular ecological methods are widely used to rapidly and efficiently observe the microbial composition in food fields [4]. Earlier, numerous studies used metagenome sequencing techniques based on 16S rRNA gene fragments to analyze the microbiome in traditional fermented foods [18,21,24]. However, the 16S rRNA gene provides low taxonomic resolution for some species [18]. Some LAB, such as *Lactiplantibacillus* species, *Latilactobacillus* species, and *W. cibaria*/*W. confusa*, which are mainly involved in vegetable or fish fermentation, were not distinguished by this method at the species level [18]. In contrast, MALDI-TOF MS-based ribosomal protein accurately identified these species [18]. Therefore, in this study, MALDI-TOF MS was evaluated as a high-throughput method for identifying microorganisms isolated from gajami-sikhae.

A total of 4824 isolates were obtained during the fermentation of gajami-sikhae. The accuracy of MALDI-TOF MS mainly depends on the reference database [25]. Before identifying the isolates, 10 reference strains mainly involved in gajami-sikhae fermentation were analyzed using the bioTyper database. All reference strains were identified with score values of 2.0 or higher (data not shown). The mass profiles of the isolates were compared to the reference spectra in the database, and then the isolates were identified at the species level based on the given score values. In addition, 4824 isolates of gajami-sikhae were identified as belonging to various genera such as *Bacillus*, *Enterobacter*, *Enterococcus*, *Lactiplantibacillus*, *Lactobacillus*, *Latilactobacillus*, *Levilactobacillus*, *Lactococcus*, *Leuconostoc*, *Pediococcus*, and *Weissella*. This finding resulted in 3805 isolates (78.88%) with a score of ≥ 2.000, corresponding to highly probable species identification (Table 1). This suggests that the MALDI-TOF MS method can identify most isolates related to gajami-sikhae fermentation. Furthermore, 1019 isolates (21.12%) delivered scores between 1.700 and 2.000, corresponding to probable genus identification (Table 2). While low identification scores (<2.000) may have different causes, such as species not present in the reference databases, a low number of representative isolates of a given species, or problems in sample preparation, they could reveal other LAB species. Therefore, strains not identified at the species level by MALDI-TOF MS should be further analyzed using housekeeping genes such as *pheS* and *rpoB* genes.

### 3.3. Bacterial Community Dynamics during Fermentation

In previous studies, microbial communities were analyzed using culture-dependent (MALDI-TOF MS) and culture-independent (metagenome sequencing) approaches [18,26,27]. Both identification systems produced almost identical results. However, MALDI-TOF MS could not identify microorganisms absent from the reference databases. Although metagenome sequencing could not accurately identify some closely related species at the species level, this approach allowed for the detection of higher biodiversity than the MALDI-TOF MS. Both approaches provided complementary information by producing a comprehensive view of the microbial ecology in environmental or food samples.

Although gajami-sikhae is a very intriguing traditional Korean fermented fish, there is little information about its microbial composition compared to other fermented foods such as kimchi and jeotgal. Moreover, Kim et al. (2014) is the only study that has identified the composition of microorganisms in different gajami-sikhae samples using pyrosequencing analysis [19]. However, the study did not report any microbial community change in gajami-sikhae during fermentation. Since gajami-sikhae is fermented in an unsterilized natural environment, leading to the growth of various microorganisms, it is necessary to investigate changes in the microbial community to standardize the quality of gajami-sikhae. Therefore, we used MALDI-TOF MS to identify the microbial community in gajami-sikhae based on the effect of varying fermentation conditions.

The most dominant genera were found to be *Latilactobacillus* (45.23%), *Leuconostoc* (12.89%), and *Weissella* (13.83%), which were present at all fermentation temperatures (Appendix A). The other major genera present include *Lactiplantibacillus* (13.70%) and *Levilactobacillus* (13.33%). This result was consistent with the findings of the previous study that confirmed the microbial community composition in gajami-sikhae using pyrosequencing [19]. The species, including *L*. *sakei*, *L*. *plantarum*, *L*. *brevis*, *Leu*. *mesenteroides*, and *W*. *koreensis,* were identified with high abundance during fermentation at different temperatures. The predominant species in the microbial community of gajami-sikhae belonged to lactic acid bacterial species responsible for variations in the sensory qualities of other Korean fermented foods, such as kimchi and jeotgal [28,29]. The discovery that the microbial community in gajami-sikhae is similar to that in kimchi for fermenting vegetables is probably because of the similar production methods of sikhae and kimchi [19].

Microbial communities demonstrated a similar pattern in gajami-sikhae fermented at 5 °C and 10 °C, and a similar pattern for fermentation at 15 °C and 20 °C. Figure 3 and Appendix A represent the microbial communities identified at the species and genus level, respectively. As shown in Figure 3, *L*. *sakei*, *Leu*. *mesenteroides*, *Leu*. *gelidum*, *Leu*. *citreum*, and *W*. *koreensis* were abundant at the beginning of fermentation (Sample C), suggesting that these species were a major component of the microbial community of the raw materials. *Weissella* species, such as *W*. *koreensis* and *W*. *cibaria,* were isolated in samples fermented at all temperatures. These species were found only at the early stage of fermentation because their growth was affected by acid [30]. *Leu*. *mesenteroides*, major LAB in fermented vegetables, are mainly used as a starter in commercial food fermentation because they produce mannitol. Mannitol is a naturally occurring 6-carbon diabetic polyol that provides a refreshing taste [31]. In the low-temperatures fermented samples, *Leu*. *mesenteroides* existed until the later stage of fermentation, but in the high-temperatures fermented samples, it decreased rapidly as the fermentation process continued. Also, these findings showed that *L*. *sakei* predominated the early stage of gajami-sikhae fermentation in all temperatures. This species is the dominant species in the microbial community of fermented fish and may play some role in the fermentation process [24]. During the fermentation process, *L*. *sakei* increased and predominated the later stage of gajami-sikhae fermentation at 5 °C and 10 °C. However, *L*. *brevis* and *Lactiplantibacillus plantarum* replaced it at 15°C and 20°C. *L*. *plantarum* and *L*. *brevis* increased in samples fermented at 15 °C and 20 °C and stabilized in the later fermentation stage, becoming the only dominant species. At low temperatures (5 °C and 10 °C), this species was not identified. A rapid increase in acidity and establishment of anaerobic conditions toward the later stage of fermentation is favorable for the growth of *Lactiplantibacillus* species since they adapt well to anaerobic and highly acidic conditions [20]. Thus, the relative abundance of *L*. *plantarum* and *L*. *brevis* reached 100% at the end of fermentation in the gajami-sikhae fermented at 20 °C.

### 3.4. Relationship between Environmental Factors and the Microbial Community

The correlations between the relative abundance of species involved in the fermentation of gajami-sikhae and factors such as the pH, acidity, fermentation temperature, and fermentation period were analyzed to determine the effect of these factors on the microbial composition (Figure 4). *L*. *sakei* was identified as the major species of the microflora in gajami-sikhae. Previously, *L*. *sakei* has been reported as the major microorganism found in Korean fermented foods such as kimchi [18,20]. This species showed a negative correlation with fermentation temperature (Pearson coeffect *r* = −0.505, *p* = 1.384 × 10^−3^), with a tendency to decrease when the temperature increased. This finding was consistent with previous reports that *L*. *sakei* adapts well at low temperatures [18]. In contrast, *L*. *plantarum* (Pearson coeffect *r* = 0.614, *p* = 5.914 × 10^−5^) and *L*. *brevis* (Pearson coeffect *r* = 0.523, *p* = 8.663 × 10^−4^) demonstrated a positive correlation with fermentation temperature. Also, these LAB species showed a strong positive correlation with acidity (*L*. *plantarum*, Pearson coeffect *r* = 0.819, *p* = 1.986 × 10^−9^; *L*. *brevis*, Pearson coeffect *r* = 0.600, *p* = 8.589 × 10^−5^). Therefore, *L*. *plantarum* and *L*. *brevis* are well adapted to acidic environments and high temperatures [32].

*Leu*. *mesenteroides and W*. *koreensis* are often isolated from meat or vegetable products fermented at low temperatures and under weak acidic conditions [33,34]. *Leu*. *mesenteroides* and *W*. *koreensis* had a negative correlation with acidity (*Leu*. *mesenteroides*, Pearson coeffect *r* = −0.614, *p* = 5.914 × 10^−5^; *W*. *koreensis*, Pearson coeffect *r* = −0.647, *p* = 3.197 × 10^−5^) and fermentation period (*Leu*. *mesenteroides*, Pearson coeffect *r* = −0.618, *p* = 5.914 × 10^−5^; *W*. *koreensis*, Pearson coeffect *r* = −0.470, *p* = 2.791 × 10^−3^). They demonstrated a tendency to decrease as the acidity and fermentation period increased. *Leu*. *gelidum* (Pearson coeffect *r* = −0.748, *p* = 2.463 × 10^−7^) and *W*. *koreensis* (Pearson coeffect *r* = −0.501, *p* = 1.409 × 10^−3^) showed a negative correlation with fermentation temperature. Therefore, the abundance of *Leu*. *gelidum* and *W*. *koreensis* decreased as the fermentation temperature increased, whereas that of *L*. *plantarum* and *L*. *brevis* increased. These results suggest that gajami-sikhae should be fermented at low temperatures to increase the proliferation of *Leuconostoc* and *Weissella* species. In addition, *Leuconostoc* and *Weissella* species are beneficial bacteria that provide the flavor of fermented foods [31].

The correlation analysis between the major species and environmental factors showed that *Leu*. *mesenteroides* and *W*. *koreensis* did not adapt well in an acidic environment, whereas *L*. *plantarum* and *L*. *brevis* adapted well. In addition, *L*. *sakei*, *Leu*. *gelidum*, and *W*. *koreensis* grew well at low temperatures (5 °C and 10 °C), whereas *L*. *plantarum* and *L*. *brevis* grew well at high temperatures (15 °C and 20 °C). This finding corresponds to the previous studies, which showed that heterofermentative LAB such as *Leu*. *mesenteroides* predominate under weaker acidic and lower anaerobic conditions during fermentation. Furthermore, heterofermentative LAB, such as *L*. *plantarum* (facultative heterofermentative species) and *L*. *brevis* (obligate heterofermentative species), become dominant as food fermentation conditions change to more anaerobic and acidic conditions [21,31,35].

Since only one batch was used in this study, variation between batches cannot be expected. According to a previous study, pyrosequencing data showed variation in the microbial compositions between gajami-sikhae samples from eight different manufacturers; the microbial compositions of two out of the eight gajami-sikhae samples were distinct from those of the rest [19]. In another study, 88 samples of kimchi, which has a similar microbial composition to gajami-sikhae, were examined to identify their microbial communities, and it was reported that there was little variation in microbial communities due to the shared ingredients and standardized manufacturing process [36]. Further research will be needed to observe the batch-to-batch variation in the microbial community in gajami-sikhae.

## 4. Conclusions

In this study, for the first time, we analyzed changes in the lactic acid bacterial community in gajami-sikhae using MALDI-TOF MS. Our studies accurately identified the LAB in gajami-sikhae at the species or genus level using MALDI-TOF MS and observed the dominant species. *L*. *sakei*, *L*. *plantarum*, *L*. *brevis*, *Leu*. *mesenteroides*, and *W*. *koreensis* were the key fermentative microbes in gajami-sikhae fermentation. The dominant species differed depending on the fermentation temperature and period, suggesting that the fermentation temperature and period are important indices determining the quality of gajami-sikhae. These results provide information on the fermentation conditions (fermentation temperature and period) of gajami-sikhae. Also, the information provided in this study will be useful in developing effective strategies for selecting bacterial strains. Future research should focus on sensory analysis and volatility profile analysis to improve the quality of gajami-sikhae.

## Figures and Tables

**Figure 1 foods-11-00909-f001:**
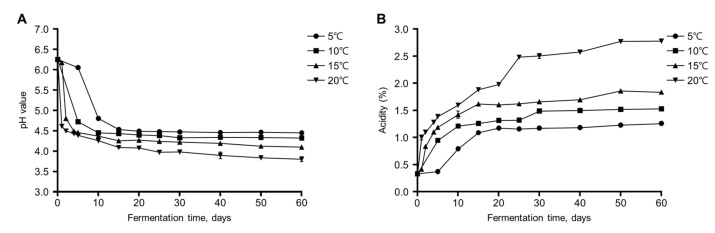
Changes in the pH and acidity of gajami-sikhae samples during the fermentation at different fermentation temperatures: (**A**) pH profiles; (**B**) acidity profiles. Error bars represent the mean ± standard deviation.

**Figure 2 foods-11-00909-f002:**
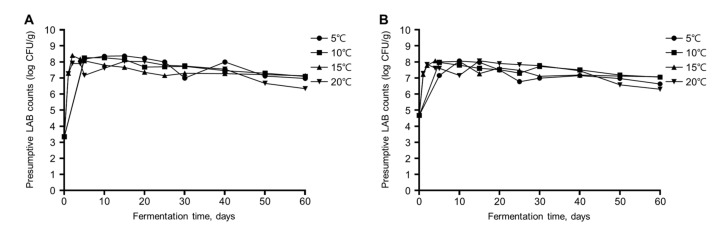
Changes in presumptive LAB counts (log CFU/g) of gajami-sikhae samples during fermentation at different fermentation temperatures: (**A**) presumptive LAB count incubated at 20 °C; (**B**) presumptive LAB count incubated at 30 °C.

**Figure 3 foods-11-00909-f003:**
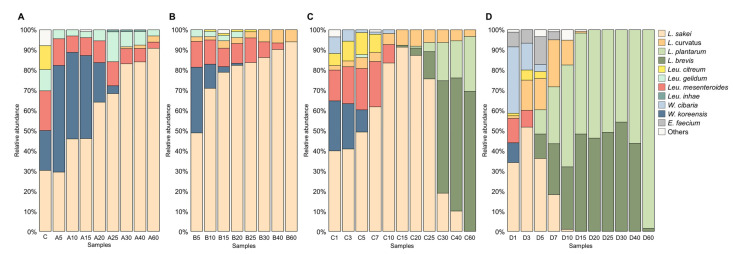
Changes in the LAB communities during the fermentation at (**A**) 5 °C, (**B**) 10 °C, (**C**) 15 °C, and (**D**) 20 °C. The graphs were generated by considering only the 3805 cultures identified at the species level (score ≥ 2.000). “Others” indicates species with a prevalence of 0.15%, including *P*. *pentosaceus*, *Lc*. *lactis*, *W*. *viridescens*, *W*. *hellenica*, *Leu*. *lactis*, and *W*. *kandleri*.

**Figure 4 foods-11-00909-f004:**
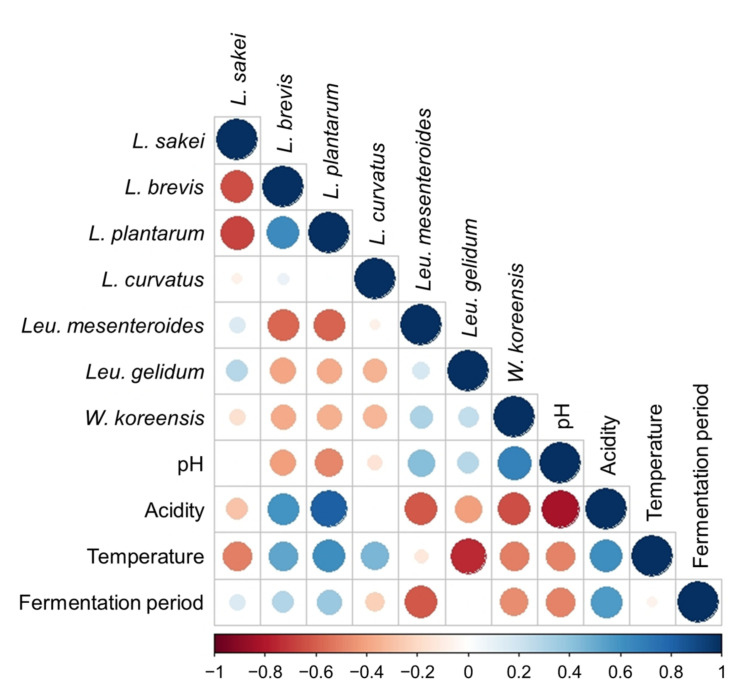
Pearson correlations calculated for the environmental factors (pH, acidity, fermentation temperature, and fermentation period) and relative abundance of major lactic acid bacterial species. The blue and red colors correspond to positive and negative correlations, respectively. The circle size and color intensity are proportional to the correlation coefficient.

**Table 1 foods-11-00909-t001:** Number of isolates with correct identification to the species level by MALDI-TOF MS.

Species (No. of Isolates)	No. of Isolates with Results
*Bacillus pumilus* (1)	0 (0%)
*Bacillus subtilis* (1)	0 (0%)
*Enterobacter cowanii* (1)	0 (0%)
*Enterococcus faecium* (28)	22 (78.57%)
*Enterococcus hermanniensis* (2)	0 (0%)
*Lactiplantibacillus plantarum* (661)	504 (76.25%)
*Fructilactobacillus fructivorans* (1)	1 (100%)
*Latilactobacillus curvatus* (217)	155 (71.43%)
*Latilactobacillus sakei* (1965)	1873 (95.32%)
*Levilactobacillus brevis* (643)	563 (87.56%)
*Lactococcus lactis* (7)	7 (100%)
*Leuconostoc citreum* (118)	47 (39.83%)
*Leuconostoc gelidum* (140)	69 (49.29%)
*Leuconostoc inhae* (30)	3 (10%)
*Leuconostoc lactis* (3)	0 (0%)
*Leuconostoc mesenteroides* (329)	238 (72.34%)
*Leuconostoc pseudomesenteroides* (2)	0 (0%)
*Pediococcus acidilactici* (1)	0 (0%)
*Pediococcus pentosaceus* (7)	7 (100%)
*Weissella cibaria* (52)	52 (100%)
*Weissella hellinica* (6)	1 (16.67%)
*Weissella kandleri* (3)	2 (66.67%)
*Weissella koreensis* (599)	256 (42.74%)
*Weissella viridescens* (7)	5 (71.43%)
Total (4824)	3805 (78.88%)

**Table 2 foods-11-00909-t002:** Number of isolates with correct identification to the genus level by MALDI-TOF MS.

Genus (No. of Isolates)	No. of Isolates with Results
*Bacillus pumilus* (1)	1 (100%)
*Bacillus subtilis* (1)	1 (100%)
*Enterobacter cowanii* (1)	1 (100%)
*Enterococcus faecium* (28)	6 (21.43%)
*Enterococcus hermanniensis* (2)	2 (100%)
*Lactiplantibacillus plantarum* (661)	157 (23.75%)
*Fructilactobacillus fructivorans* (1)	0 (0%)
*Latilactobacillus curvatus* (217)	62 (28.57%)
*Latilactobacillus sakei* (1965)	92 (4.68%)
*Levilactobacillus brevis* (643)	80 (12.44%)
*Lactococcus lactis* (7)	0 (0%)
*Leuconostoc citreum* (118)	71 (60.17%)
*Leuconostoc gelidum* (140)	71 (50.71%)
*Leuconostoc inhae* (30)	27 (90%)
*Leuconostoc lactis* (3)	3 (100%)
*Leuconostoc mesenteroides* (329)	91 (27.66%)
*Leuconostoc pseudomesenteroides* (2)	2 (100%)
*Pediococcus acidilactici* (1)	1 (100%)
*Pediococcus pentosaceus* (7)	0 (0%)
*Weissella cibaria* (52)	0 (0%)
*Weissella hellinica* (6)	5 (83.33%)
*Weissella kandleri* (3)	1 (33.33%)
*Weissella koreensis* (599)	343 (57.26%)
*Weissella viridescens* (7)	2 (28.57%)
Total (4824)	1019 (21.12%)

## Data Availability

The data presented in this study are available on request from the corresponding author.

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
