# Peer review of "Diversity of a Lactic Acid Bacterial Community during Fermentation of Gajami-Sikhae, a Traditional Korean Fermented Fish, as Determined by Matrix-Assisted Laser Desorption/Ionization Time-of-Flight Mass Spectrometry"

_foods, 2022, doi:10.3390/foods11070909_

Round 1

Reviewer 1 Report

The focus of the present paper deals is the characterization of the dominant lactic microflora of gajami-sikhae, a Korean traditional fermented fish-based dish and four different temperature levels were considered.

A cultural approach was adopted and MALDI TOF was used for strains’ identification.

A large number of cultures was isolated, but as matter of fact a unique spontaneous fermentation was monitored, and this in a way jeopardizes outcomes.

Apart from this, the manuscript needs to be revised by an English native speaker.

Specifically:

Line 15. In the abstract, isolated cultures were 5801. In results they pass to4824 (line 481) without any explanation.

Line 15. Authors speak in first person throughout the manuscript. This is something I do not prefer for scientific publications, but of course this is just an opinion of mine.

Line 22. L. plantarum and L. brevis are known to produce off-flavor in fermented vegetables? I’m not agree and anyhow this concept is no further commented…

Line 24. I’m not sure that this sentence is justified as is. Authors could state that their study could allow the selection of an adjunct culture to control the gajami-sikhae fermentation. That’s all.

Lines 49-70. This speculation is completely off topic. Authors should focus on the circumstance that gajami-sikhae microbiome has been investigated just once by pyrosequencing and that the effect of temperature on this fermentation has never been investigated before.

Lines 67-68. A reference is mandatory

Lines 69-71. Sure?

Lines 79-81. gajami-sikhae was purchased or prepared? Please clarify

Line 83. Please add plastic bowls characteristics and size

Line 97. Introduce the abbreviation LAB for Lactic Acid Bacteria

Lines 102-103. All colonies from countable plates were picked?

Figure 1. What is reported in panel D? Moreover, at the onset of fermentation pH and TTA change with temperatures, but not the CFU counts. I would prefer that data are reported on a table and that the statistical significance of the differences is reported.

Table 1 is useless

Table 2. Column ≤1.699 is useless, on the other hand strains identified at a score of 1.700–1.999 should not be reported as species!!! please split data in two tables

Figure 2. Even in this case I would expect a graph generated by considering only for the 3805 cultures identified at species level, but actually I cannot be sure about this. Another aspect that really puzzles me is: where sample A1, A3, B1, B3 are? In figure 1 counts higher than 4 Log CHU/g are reported… And what is C and why it is reported in figure 1?

Author Response

Response to Reviewer 1 Comments

The focus of the present paper deals is the characterization of the dominant lactic microflora of gajami-sikhae, a Korean traditional fermented fish-based dish and four different temperature levels were considered. A cultural approach was adopted and MALDI TOF was used for strains’ identification. A large number of cultures was isolated, but as matter of fact a unique spontaneous fermentation was monitored, and this in a way jeopardizes outcomes.

Response: Thank you for your critical review. As you mentioned, this study analyzed the microbial community for spontaneous fermentation under unique fermentation condition. However, this is first study to identify the microbial community in gajami-sikhae during the fermentation at different fermentation temperatures. In further research, it will be necessary to analyze various fermentation conditions for gajami-sikhae. As you recommended, we added the sentence to the results and discussion section in lines 324-332 as follows:

Lines 324-332: Since only one batch was used in this study, variation between batches cannot be expected. According to a previous study, pyrosequencing data showed variation of microbial compositions between gajami-sikhae samples from eight different manufacturers; the microbial composition of two out of eight gajami-sikhae samples were distinct from those of the rest [19]. In another study, 88 kimchi samples, which had similar microbial composition to gajami-sikhae, were identified for microbial communities and reported that there was little variation in microbial communities due to the shared ingredients and standardized manufacturing process [36]. Further researches will be need to observe the batch-to-batch variation of microbial community in gajami-sikhae.

Apart from this, the manuscript needs to be revised by an English native speaker.

Response: The original manuscript was edited professionally by Enago Editing Services (reference number: INQ-915916922). If the English level of the entire revised manuscript does not meet the level required by the journal, we will re-edit it to a native speaker.

Specifically:

Line 15. In the abstract, isolated cultures were 5801. In results they pass to 4824 (line 481) without any explanation.

Response: Isolates cultures were 4824. 5801 isolated cultures in the abstract were wrong. As you recommended, we revised the sentence in line 15 as follows:

Line 15: A total of 4,824 isolates were isolated from the fermented gajami-sikhae.

Line 15. Authors speak in first person throughout the manuscript. This is something I do not prefer for scientific publications, but of course this is just an opinion of mine.

Response: As you recommended, we revised the sentence in lines 15-17, 24-25, 103-104, 115-119, 119, 138, 173, 197, 201-203, 262, and 341 as follows:

Lines 15-17: A total of 4,824 isolates were isolated from the fermented gajami-sikhae. These findings indicated Latilactobacillus, Lactiplantibacillus, Levilactobacillus, Weissella, and Leuconostoc were the dominant genera during fermentation.

Lines 24-25: This study could allow the selection of an adjunct culture to control the gajami-sikhae fermentation.

Lines 103-104: For isolation of LAB, 25 g of gajami-sikhae sample and 225 ml of sterilized phosphate-buffered saline (PBS) was placed in a sterile stomacher bag (Seward Limited, London, UK).

Lines 115-119: Subsequently, 1 µl HCCA matrix solution containing 10 mg/ml α-cyano-4-hydroxycinnamic acid (CHCA) (Bruker Daltonics) in acetonitrile, water, and trifluoracetic acid (50:47.5:2.5 (v/v/v)) was added to the spot and dried again. The polished steel MALDI target plate was introduced into the Microflex LT bench-top MALDI-TOF mass spectrometer (Bruker Daltonics).

Line 119: The mass spectra of isolates were identified

Line 138: fermented period was determined

Line 173: The LAB population of gajami-sikhae was estimated by plate counting on MRS agar.

Line 197: A total of 4,824 isolates were obtained during the fermentation of gajami-sikhae

Lines 201-203: The mass profiles of the isolates were compared to the reference spectra in the database, and then the isolates were identified at the species level

Line 262: these findings

Line 341: These results provide information

Line 22. L. plantarum and L. brevis are known to produce off-flavor in fermented vegetables? I’m not agree and anyhow this concept is no further commented…

Response: As you recommended, we revised the sentence in lines 21-22 as follows:

Lines 21-22: The relative abundance of L. plantarum and L. brevis reached 100% at the later fermentation stage at 20°C

Line 24. I’m not sure that this sentence is justified as is. Authors could state that their study could allow the selection of an adjunct culture to control the gajami-sikhae fermentation. That’s all.

Response: As you recommended, we revised the sentence in lines 24-25 as follows:

Lines 24-25: This study could allow the selection of an adjunct culture to control the gajami-sikhae fermentation.

Lines 49-70. This speculation is completely off topic. Authors should focus on the circumstance that gajami-sikhae microbiome has been investigated just once by pyrosequencing and that the effect of temperature on this fermentation has never been investigated before.

Response: As reviewers 1 and 2 recommended, we revised the sentence in lines 53-56, 58-61, 66-70, and 71-77 as follows:

Lines 53-56: Additionally, there are approaches to investigate viable microbial communities using metagenetics (e.g., after total RNA extraction and reverse PCR or through the use ethidium monoazide (EMA) treatment-PCR to only amplify DNA from viable cells)

Lines 58-61: Culture-based approaches are still widely used to analyze microbial communities. The development of new tools, such as Matrix-assisted laser desorption/ionization time-of-flight mass spectrometry (MALDI-TOF MS), has allowed to reduce time to detection as compared to conventional culture-based methods [11].

Lines 66-70: MALDI-TOF MS is superior to the 16S rRNA gene for taxonomic resolution at the species or subspecies level for some closely related species such as Lacticaseibacillus casei/L. paracasei and Lactobacillus acidophilus group species [14–17]. However, MALDI-TOF MS relies on a spectral database, only species present in the database can be identified.

Lines 71-77: Many studies addressing its use in experimental approaches related to pathogenic bacteria have been published [12,13]. Some studies have applied MALDI-TOF MS technology to observe the changes in the culturable microbial community in fermented foods, but there is no study examining the microbial community of gajami-sikhae [18]. Moreover, although the microbial community of gajami-sikhae has been investigated by pyrosequencing [19], the effect of temperatures on this fermentation has not yet been investigated.

Lines 67-68. A reference is mandatory

Response: As you recommended, we added references in lines 66-69 as follows:

Lines 66-69: Moreover, MALDI-TOF MS is superior to the 16S rRNA gene for taxonomic resolution at the species or subspecies level for some closely related species such as Lacticaseibacillus casei/L. paracasei and Lactobacillus acidophilus group species [14–17].

Lines 69-71. Sure?

Response: Some studies have applied MALDI-TOF MS technology to observe the changes in the culturable microbial community in fermented foods, but there is no study examining the microbial community of gajami-sikhae. As you recommended, we revised the sentence in lines 72-77 as follows:

Lines 72-77: Some studies have applied MALDI-TOF MS technology to observe the changes in the culturable microbial community in fermented foods, but there is no study examining the microbial community of gajami-sikhae [18]. Moreover, although the microbial community of gajami-sikhae has been investigated by pyrosequencing [19], the effect of temperatures on this fermentation has not yet been investigated.

Lines 79-81. gajami-sikhae was purchased or prepared? Please clarify

Response: Gajami-sikhae was purchased. As you recommended, we revised the sentence in lines 85-88 as follows:

Lines 85-88: A gajami-sikhae sample was purchased from the traditional manufacturers in Korean in December 2020. Sample prepared using the traditional method that the salted flatfish (Glyptocephalus stelleri) was mixed with radish, red pepper powder, boiled millet (Setaria italica), chopped garlic, and NaCl was purchased [19].

Line 83. Please add plastic bowls characteristics and size

Response: As you recommended, we added plastic bowls characteristic and size in lines 89-90 as follows:

Lines 89-90: 20°C for 60 days in the poly cyclohexane-1,4-dimethylene terephthalate plastic bowl (20×11 cm).

Line 97. Introduce the abbreviation LAB for Lactic Acid Bacteria

Response: As you recommended, we introduced the abbreviation LAB for lactic acid bacteria in the entire manuscript.

Lines 102-103. All colonies from countable plates were picked?

Response: All colonies from countable plates were picked. We revised the sentence in lines 108-110 as follows:

Lines 108-110: All colonies from countable plates with lactic acid bacterial growth between 30 and 300 CFU/plate on MRS agar were picked.

Figure 1. What is reported in panel D? Moreover, at the onset of fermentation pH and TTA change with temperatures, but not the CFU counts. I would prefer that data are reported on a table and that the statistical significance of the differences is reported.

Response: Figure 1C and 1D are figure showing the changes in presumptive LAB counts (log CFU/g) of gajami-sikhae samples during the fermentation at different fermented temperatures. As mentioned in lines 106-108, Figure 1D is a graph showing presumptive LAB counts incubated at 20°C. As reviewers 1 and 2 recommended, we split the figures for pH/acidity and presumptive LAB counts. And we newly analyzed the statistical significance of the pH and acidity. Since the LAB count was not repeated, statistical significance could not be obtained.

Lines 134-137: The statistical analysis for pH and acidity values was performed using R v.4.1.0. Significant differences (p < 0.05) between the sample means were determined by Duncan’s multiple range test.

We newly added statistical significance of differences for pH and acidity to Supplementary Table S2.

Table 1 is useless

Response: As you recommended, we deleted Table 1 and revised the sentence in lines 200-201 as follows:

Lines 200-201: All reference strains were identified with score values of 2.0 or higher (data not shown).

Table 2. Column ≤1.699 is useless, on the other hand strains identified at a score of 1.700–1.999 should not be reported as species!!! please split data in two tables

Response: As you recommended, we deleted a column ≤1.699 and split the data in two tables

Line 216: Table 1. Number of isolates correct identification to the species level by MALDI-TOF MS

Line 217: Table 2. Number of isolates correct identification to the genus level by MALDI-TOF MS

Figure 2. Even in this case I would expect a graph generated by considering only for the 3805 cultures identified at species level, but actually I cannot be sure about this. Another aspect that really puzzles me is: where sample A1, A3, B1, B3 are? In figure 1 counts higher than 4 Log CHU/g are reported… And what is C and why it is reported in figure 1?

Response: As reviewers 1 and 2 recommended, we newly generated a graph considering only for the 3,805 cultures identified at the species level. Samples A1, A3, B1, and B3 are absent. As described in Table S1, sampling of gajami-sikhae fermented at low temperatures (5°C and 10°C) was performed at 5-days intervals. For gajami-sikhae fermented at high temperatures (15°C and 20°C), sampling was performed at 2-days interval during the first 7-days, after which it was performed at 5-days or 10-days interval. In the previous studies, when fermenting at a low temperature, samples were sampled at interval of five or seven days (Lee et al., 2018). Figure 1C and D show viable cell counts. As reviewers 1 and 2 recommended, we split the figures for pH/acidity and presumptive LAB counts. And we newly analyzed the statistical significance of the pH and acidity to Table S2.

Lines 250-252: Figure 3 and Figure S1 represent the microbial communities identified at the species and genus level, respectively.

Lines 276-277: Changes in the LAB communities during the fermentation at (A) 5°C, (B) 10°C, (C) 15°C, and (D) 20°C. A graph generated by considering only for the 3,805 cultures identified at the species level (score ≥ 2.000).

We newly added a graph generated by considering only for the 1,019 cultures identified at the genus level to Supplementary Figure S1.

Reviewer 2 Report

This paper by Kim et al. entitled “Diversity of a lactic acid bacterial community during fermentation of gajami-sikhae, a Korean traditional fermented fish, by matrix-assisted laser desorption/ionization time-of-flight mass spectrometry” deals with the diversity of lactic acid bacteria (LAB) encountered in Gajami-sikhae, a Korean traditional fermented fish. Overall, this study brings relevant and new knowledge concerning LAB diversity in this naturally fermented food product which was evaluated using a culture-dependent approach combined with MALDI-ToF MS for LAB identification, on a very large number of isolates collected throughout fermentation which was conducted at different temperatures.

Major comments
My first comment concerns the English grammar, style and spelling, and overall writing of the manuscript that clearly deserves significant improvements (see minor comments for details). 

In different parts of the manuscript (L49-60; L205-214), the authors are discussing the limitations of culture-independent approaches (eg, metagenetics) to justify the use of MALDI-ToF MS as a tool to identify isolates  collected in the present study. Instead of opposing the two approaches, the authors should rather say that both approaches are complementary as they both have drawbacks. 
Indeed, culture-dependent approaches have limitations, eg, not all microorganisms are cultivable on a single medium, can be in viable but not cultivable state. In general, only a low number of samples can be analyzed using culture-dependent approaches as it is quite laborious. Concerning the use of MALDI-TOF MS for microbial identification, its identification accuracy notably depends on the implemented database (both at the species and intra-species level), Therefpre, It would have been useful to perform a clustering of MS spectra and to sequence a gene of taxonomic interest for LAB identification (eg, pheS, rpoB) in representative isolates of each cluster for confirming the identification provided. Furthermore, ~20% of collected isolates were not identifed to the species level so such an approach would have been useful to apply. While low identification scores may have different causes (species not present in the database, low number of representative isolates of a given species, problems in sample preparation), it could have revealed other LAB species. Such limitations and perspectives should be underlined and discussed by the authors. In addition, in Figure 2, the percentage of isolates not identified to the species level should be included in the barplot.

In figure 3, the authors show a Pearson correlation matrix between LAB species and other factors (temperature, pH, acidity, fermentation time). First, only significant correlations should be shown and P values should be corrected with Benjamini-Hochberg correction. Based on this new analysis, the authors should correct this part of the manuscript.

Finally, unless I misunderstood, the authors "only" analyzed 1 sample x time point x fermentation temperature all coming from the same production batch which was produced at a laboratory scale. It raises the question of the representativity of the obtained results. Such a limitation has to be clearly underlined in the results and discussion section.

Minor comments
Line 12-13: Please rephrase "The aim of this study was to investigate the diversity and dynamics of lactic acid bacteria in gajami-sikhae"
L13: replace "the cultivable microbial community" by "the diversity and dynamics of lactic acid bacteria", microbial diversity was analyzed after plating on one single medium, ie, MRS, which is a semi-selective medium dedicated to LAB. Other microorganisms, not growing onto MRS, may be present. 
L15: "We isolated 5,801 strains": the word "strains" is not appropriate as many isolates may correspond to a same strain on a plate, please rephrase
L27: "Fermented temperature", use instead "Fermentation temperature"
L35: "health benefits such as anticancer": is it scientifically proven? If not, replace by "potential health benefits"
L40: "the main strains", replace by "the main taxa"
L42: "The quality of fermented foods correlated with", LAB are not the only microorganisms that contribute to the quality of fermented foods. Please rephrase.
L43: "they produce bacteriocins"
L45: "by fermented conditions", not clear, "by fermentation conditions"
L49-53: PCR-DDGE is not anymore used in microbial ecology, please delete. 
L50: Culture-based approaches are still widely used to analyze microbial communities. The authors of the present study applied such an approach.
L56: "to detect", "to study"
L58: the species information, please rephrase, not clear
L59-60: There are approaches to investigate viable microbial communities using metagenetics, eg, after total RNA extraction and reverse PCR or through the use EMA PCR to only amplify DNA from viable cells, please correct and rephrase
L66-67: "MALDI-TOF MS is superior to the 16S rRNA gene sequencing", such an affirmation is not true, sometimes it is sometimes it is not. As reflected in the present study, >20% if collected isolated could only be assigned to the genus level. 16S rRNA gene sequencing would have certainly been able to assign many of these isolates to the species level. Moreover, as MALDI-TOF MS relies on a spectral database, only species present in the database can be identified.
L68-69: "its great application", please rephrase
L86: please change title "pH and acidity measurements"
L91: Why pH 8.2 was used for titration? Initial pH of fish is 6.2.
L115: please indicate the number of LAB species (and number of spectra) present in the database
Figure 1 C and D: change name of y-axis by "presumptive LAB counts", counts are expressed in CFU/ml, it should be CFU/g.
L161-162: sentence not clear, please rephrase
L166: "the number of LAB cells", please replace by "LAB counts"
L169-170: sentence not clear, please rephrase
L166-171: counts are expressed in CFU/ml, they should be expressed in CFU/g (25 g were analyzed!)
L184: "4,824 isolates": contradictory with what is indicated in the abstract (L15: 5,801)
L195-196: given the fact that around 20% isolates could not be identified, it cannot be stated that MALDI-TOF MS is sufficient to identify LAB associated to the studied product.
L197-198: please rephrase, MALDI-TOF MS does not detect living cells. The culture-dependent approach does.
L220: species nam should be cited in full the first time and then only the first letter of the genus should be written, Leu. is not correct. Correct throughout the manuscript
Table 2: "Bacillus pumilus"
Table 2: the correct name of "Lactobacillus homohiochii " is "Fructilactobacillus fructivorans" 
L231-233: sentence not clear, please rephrase
L236-239: sentence not clear, please rephrase
L242: L. sakei was increased, please rephrase
L251-253: contradictory with the previous sentence. Please rephrase and correct
L265-268: "L. sakei 266 adapted well at low temperatures [28], unlike L. sakei,"contradictory with the previous statement. Please rephrase and correct
L269: "these strains", "these LAB species"
L271-272: sentence not clear, which species ?
L279: "weaker", "weak"
L289-290: not clear, which bacterium? the previous sentence mention Weisella and Leuconostoc species
L295: use the past tense, cite a reference showing this.
Be more specific, what is a low and high temperature?
L299: sentence not clear, L. plantarum is a facultative heterofermentative while L. brevis is an obligate heterofermentative species 
L304: please rephrase, 20% isolates could not be assigned
L305-306: not clear
L310-311: what do you mean by "standardization" ? Be more specific please.
Conclusion : please identify future perspectives of the present study including sensorial analysis, volatile profile analysis, strain selection etc...

Author Response

Response to Reviewer 2 Comments

This paper by Kim et al. entitled “Diversity of a lactic acid bacterial community during fermentation of gajami-sikhae, a Korean traditional fermented fish, by matrix-assisted laser desorption/ionization time-of-flight mass spectrometry” deals with the diversity of lactic acid bacteria (LAB) encountered in Gajami-sikhae, a Korean traditional fermented fish. Overall, this study brings relevant and new knowledge concerning LAB diversity in this naturally fermented food product which was evaluated using a culture-dependent approach combined with MALDI-ToF MS for LAB identification, on a very large number of isolates collected throughout fermentation which was conducted at different temperatures.

Major comments

My first comment concerns the English grammar, style and spelling, and overall writing of the manuscript that clearly deserves significant improvements (see minor comments for details).

Response: The original manuscript was edited professionally by Enago Editing Services (reference number: INQ-915916922). If the English level of the entire revised manuscript does not meet the level required by the journal, we will re-edit it to a native speaker.

In different parts of the manuscript (L49-60; L205-214), the authors are discussing the limitations of culture-independent approaches (eg, metagenetics) to justify the use of MALDI-ToF MS as a tool to identify isolates collected in the present study. Instead of opposing the two approaches, the authors should rather say that both approaches are complementary as they both have drawbacks. Indeed, culture-dependent approaches have limitations, eg, not all microorganisms are cultivable on a single medium, can be in viable but not cultivable state. In general, only a low number of samples can be analyzed using culture-dependent approaches as it is quite laborious. Concerning the use of MALDI-TOF MS for microbial identification, its identification accuracy notably depends on the implemented database (both at the species and intra-species level).

Response: As you recommended, we added the sentence in lines 53-56 and 219-226 as follows:

Lines 53-56: Additionally, there are approaches to investigate viable microbial communities using metagenetics (e.g., after total RNA extraction and reverse PCR or through the use ethidium monoazide (EMA) treatment-PCR to only amplify DNA from viable cells)

Lines 219-226: In the previous studies, microbial communities were analyzed using culture-dependent (MALDI-TOF MS) and culture-independent approach (metagenome sequencing) [18,26,27]. Both identification systems produced almost identical results. However, MALDI-TOF MS could not identify microorganisms absent from the reference databases. Although metagenome sequencing could not accurately identify some closely related species at the species level, this approach allowed for the detection of higher biodiversity than the MALDI-TOF MS. Both approaches provided complementary information by producing comprehensive view of the microbial ecology in environmental or food samples.

Therefore, it would have been useful to perform a clustering of MS spectra and to sequence a gene of taxonomic interest for LAB identification (eg, pheS, rpoB) in representative isolates of each cluster for confirming the identification provided. Furthermore, ~20% of collected isolates were not identifed to the species level so such an approach would have been useful to apply. While low identification scores may have different causes (species not present in the database, low number of representative isolates of a given species, problems in sample preparation), it could have revealed other LAB species. Such limitations and perspectives should be underlined and discussed by the authors.

Response: As you recommended, we added the sentence in lines 210-215 as follows:

Lines 210-215: While low identification scores (< 2.000) may have different causes, such as species not present in the reference databases, low number of representative isolates of a given species, or problems in sample preparation, it could have revealed other LAB species. Therefore, strains not identified at the species level by MALDI-TOF MS should be further analyzed using housekeeping genes such as pheS and rpoB genes.

In addition, in Figure 2, the percentage of isolates not identified to the species level should be included in the barplot.

Response: As reviewers 1 and 2 recommended, we presented in Table S2 for the percentage of isolates that were not identified at the species level.

In figure 3, the authors show a Pearson correlation matrix between LAB species and other factors (temperature, pH, acidity, fermentation time). First, only significant correlations should be shown and P values should be corrected with Benjamini-Hochberg correction. Based on this new analysis, the authors should correct this part of the manuscript.

Response: As you recommended, we presented significant correlations, and P values are newly corrected with Benjamini-Hochberg correction. And, we added the sentence in lines 139-140, 286, 288-289, 291-292, 301-302, 303-304, and 305-306 as follows:

Lines 139-140: The Benjamini-Hochberg correction was used to correct the P values [22].

Line 286: Pearson coeffect r = −0.505, p = 1.383600e-03).

Lines 288-289: L. plantarum (Pearson coeffect r = 0.614, p = 5.914333e-05) and L. brevis (Pearson coeffect r = 0.523, p = 8.662500e-04)

Lines 291-292: L. plantarum, Pearson coeffect r = 0.819, p = 1.985500e-09; L. brevis, Pearson coeffect r = 0.600, p = 8.589429e-05

Lines 301-302: Leu. mesenteroides, Pearson coeffect r = −0.614, p = 5.914333e-05; W. koreensis, Pearson coeffect r = −0.647, p = 3.197333e-05

Lines 303-304: Leu. mesenteroides, Pearson coeffect r = −0.618, p = 5.914333e-05; W. koreensis, Pearson coeffect r = −0.470, p = 2.790700e-03

Lines 305-306: Leu. gelidum (Pearson coeffect r = −0.748, p = 2.462900e-07) and W. koreensis (Pearson coeffect r = −0.501, p = 1.409222e-03)

Finally, unless I misunderstood, the authors "only" analyzed 1 sample x time point x fermentation temperature all coming from the same production batch which was produced at a laboratory scale. It raises the question of the representativity of the obtained results. Such a limitation has to be clearly underlined in the results and discussion section.

Response: Thank you for your critical review. As you mentioned, this study analyzed the microbial community for spontaneous fermentation under unique fermentation condition. However, this is first study to identify the microbial community in gajami-sikhae during the fermentation at different fermentation temperatures. In further research, it will be necessary to analyze various fermentation conditions for gajami-sikhae. As you recommended, we added the sentence to the results and discussion section in lines 324-332 as follows:

Lines 324-332: Since only one batch was used in this study, variation between batches cannot be expected. According to a previous study, pyrosequencing data showed variation of microbial compositions between gajami-sikhae samples from eight different manufacturers; the microbial composition of two out of eight gajami-sikhae samples were distinct from those of the rest [19]. In another study, 88 kimchi samples, which had similar microbial composition to gajami-sikhae, were identified for microbial communities and reported that there was little variation in microbial communities due to the shared ingredients and standardized manufacturing process [36]. Further researches will be need to observe the batch-to-batch variation of microbial community in gajami-sikhae.

Minor comments

Line 12-13: Please rephrase "The aim of this study was to investigate the diversity and dynamics of lactic acid bacteria in gajami-sikhae"

Response: As you recommended, we revised the sentence in lines 12-13 as follows:

Lines 12-13: This study was the first to investigate the diversity and dynamics of lactic acid bacteria in gajami-sikhae

L13: replace "the cultivable microbial community" by "the diversity and dynamics of lactic acid bacteria", microbial diversity was analyzed after plating on one single medium, ie, MRS, which is a semi-selective medium dedicated to LAB. Other microorganisms, not growing onto MRS, may be present.

Response: As you recommended, we replaced “the cultivable microbial community” by “the diversity and dynamics of lactic acid bacteria” in lines 12-13 as follows:

Lines 12-13: This study was the first to investigate the diversity and dynamics of lactic acid bacteria in gajami-sikhae

L15: "We isolated 5,801 strains": the word "strains" is not appropriate as many isolates may correspond to a same strain on a plate, please rephrase

Response: Isolates cultures were 4824. 5801 isolated cultures in the abstract were wrong. As you recommended, we revised the word “strain” to “isolate” in line 15 as follows:

Line 15: A total of 4,824 isolates were isolated from the fermented gajami-sikhae.

L27: "Fermented temperature", use instead "Fermentation temperature"

Response: As you recommended, we changed the “Fermented temperature” to “Fermentation temperature” in line 27 as follows:

Line 27: Fermentation temperature

L35: "health benefits such as anticancer": is it scientifically proven? If not, replace by "potential health benefits"

Response: As you recommended, we revised the sentence in line 35 as follows:

Line 35: potential health benefits, such as anticancer

L40: "the main strains", replace by "the main taxa"

Response: As you recommended, we revised the sentence in line 40 as follows:

Line 40: the main taxa of fermented fish

L42: "The quality of fermented foods correlated with", LAB are not the only microorganisms that contribute to the quality of fermented foods. Please rephrase.

Response: As you recommended, we revised the sentence in lines 42-43 as follows:

Lines 42-43: The quality of fermented foods correlated to the various microorganisms that occur naturally during the fermentation process

L43: "they produce bacteriocins"

Response: As you recommended, we revised the sentence in line 43 as follows:

Line 43: they produce bacteriocins

L45: "by fermented conditions", not clear, "by fermentation conditions"

Response: As you recommended, we revised the sentence in line 45 as follows:

Line 45: by fermentation conditions

L49-53: PCR-DDGE is not anymore used in microbial ecology, please delete.

Response: As you recommended, we deleted the sentence related to PCR-DGGE.

L50: Culture-based approaches are still widely used to analyze microbial communities. The authors of the present study applied such an approach.

Response: As you recommended, we revised the sentence in lines 58-61 as follows:

Lines 58-61: Culture-based approaches are still widely used to analyze microbial communities. The development of new tools, such as Matrix-assisted laser desorption/ionization time-of-flight mass spectrometry (MALDI-TOF MS), has allowed to reduce time to detection as compared to conventional culture-based methods [11].

L56: "to detect", "to study"

Response: As you recommended, we changed “to detect” to “to study” in line 51 as follows:

Line 51: applied to study microbial communities

L58: the species information, please rephrase, not clear

Response: As you recommended, we revised the sentence in lines 56-57 as follows:

Lines 56-57: However, the species identification obtained using the metagenomic approach

L59-60: There are approaches to investigate viable microbial communities using metagenetics, eg, after total RNA extraction and reverse PCR or through the use EMA PCR to only amplify DNA from viable cells, please correct and rephrase

Response: As you recommended, we revised the sentence in lines 53-56 as follows:

Lines 53-56: Additionally, there are approaches to investigate viable microbial communities using metagenetics (e.g., after total RNA extraction and reverse PCR or through the use ethidium monoazide (EMA) treatment-PCR to only amplify DNA from viable cells)

L66-67: "MALDI-TOF MS is superior to the 16S rRNA gene sequencing", such an affirmation is not true, sometimes it is sometimes it is not. As reflected in the present study, >20% if collected isolated could only be assigned to the genus level. 16S rRNA gene sequencing would have certainly been able to assign many of these isolates to the species level. Moreover, as MALDI-TOF MS relies on a spectral database, only species present in the database can be identified.

Response: Previous studies have reported that MALDI-TOF MS is superior to the 16S rRNA gene for taxonomic resolution for some closely related species, such as Lacticaseibacillus casei/L. paracasei and Lactobacillus acidophilus group species (Huang and Huang, 2018; Anderson et al., 2014). In fact, the metagenome sequencing based on 16S rRNA gene sequence identified microbial communities in food sample down to the genus level and could not obtain accurate results at the species level (Lee et al., 2018). As you recommended, we revised the sentence in lines 66-70 as follows:

Lines 66-70: MALDI-TOF MS is superior to the 16S rRNA gene for taxonomic resolution at the species or subspecies level for some closely related species such as Lacticaseibacillus casei/L. paracasei and Lactobacillus acidophilus group species [14–17]. However, MALDI-TOF MS relies on a spectral database, only species present in the database can be identified.

L68-69: "its great application", please rephrase

Response: As you recommended, we revised the sentence in lines 71-72 as follows:

Lines 71-72: Many studies addressing its use in experimental approaches related to pathogenic bacteria have been published

L86: please change title "pH and acidity measurements"

Response: As you recommended, we changed the title in line 93 as follows:

Line 93: 2.2. pH and acidity measurements

L91: Why pH 8.2 was used for titration? Initial pH of fish is 6.2.

Response: Since the endpoint in the previously reported studies was 8.2, the pH of the food sample was titrated to 8.2 to measure acidity (Lee et al., 2018; Lee et al., 2021). We added references in line 98.

Line 98: 8.2 to measure the acidity of gajami-sikhae [20,21].

L115: please indicate the number of LAB species (and number of spectra) present in the database

Response: As you recommended, we added the number of LAB species (and number of spectra) present in the database in lines 121-122 as follows:

Lines 121-122: Bruker MSP database consists of 98 species and 236 spectra of LAB (Lactobacillus-related species).

Figure 1 C and D: change name of y-axis by "presumptive LAB counts", counts are expressed in CFU/ml, it should be CFU/g.

Response: As you recommended, we changed name of y-axis by “presumptive LAB counts (CFU/g)”.

L161-162: sentence not clear, please rephrase

Response: As you recommended, we revised the sentence in lines 171-172 as follows:

Lines 171-172: However, samples fermented at 5 °C and 10 °C did not reach acidities (1.6%–2.0%) until the end of fermentation.

L166: "the number of LAB cells", please replace by "LAB counts"

Response: As you recommended, we changed “the number of LAB cells” to “LAB counts” in line 176 as follows:

Line 176: LAB counts reached the maximum

L169-170: sentence not clear, please rephrase

Response: As you recommended, we revised the sentence in lines 178-180 as follows:

Lines 178-180: Finally, samples fermented at 15°C and 20°C reached the maximum cell counts with an average of 8.4 CFU/g and 7.9 CFU/g after three and two days of fermentation, respectively.

L166-171: counts are expressed in CFU/ml, they should be expressed in CFU/g (25 g were analyzed!)

Response: As you recommended, we changed “CFU/ml” to CFU/g in lines 176-180.

L184: "4,824 isolates": contradictory with what is indicated in the abstract (L15: 5,801)

Response: Isolates cultures were 4824. 5801 isolated cultures in the abstract were wrong. As you recommended, we revised the sentence in line 15 as follows:

Line 15: A total of 4,824 isolates were isolated from the fermented gajami-sikhae.

L195-196: given the fact that around 20% isolates could not be identified, it cannot be stated that MALDI-TOF MS is sufficient to identify LAB associated to the studied product.

Response: As you recommended, we revised the sentence in lines 210-215 as follows:

Lines 210-215: While low identification scores (< 2.000) may have different causes, such as species not present in the reference databases, low number of representative isolates of a given species, or problems in sample preparation, it could have revealed other LAB species. Therefore, strains not identified at the species level by MALDI-TOF MS should be further analyzed using housekeeping genes such as pheS and rpoB genes.

L197-198: please rephrase, MALDI-TOF MS does not detect living cells. The culture-dependent approach does.

Response: As you recommended, we deleted the sentence.

L220: species nam should be cited in full the first time and then only the first letter of the genus should be written, Leu. is not correct. Correct throughout the manuscript

Response: As you recommended, we cited the species name first time in its full and then only the first letter of the genus throughout the manuscript.

Line 41: Weissella koreensis

Lines 130-132: Leuconostoc mesenteroides KCTC 3100, Leuconostoc inhae KACC 12281, Leuconostoc gelidum KACC 12256, Weissella cibaria KCTC 3746, Weissella confusa KCTC 3499, and W. koreensis KACC 11853.

Table 2: "Bacillus pumilus"

Response: As you recommended, we revised the species name in Table 1.

Table 2: the correct name of "Lactobacillus homohiochii " is "Fructilactobacillus fructivorans"

Response: As you recommended, we changed “Lactobacillus homohiochii" to "Fructilactobacillus fructivorans” in Table 1.

L231-233: sentence not clear, please rephrase

Response: As you recommended, we revised the sentence in lines 255-257 as follows:

Lines 255-257: Weissella species, such as W. koreensis and W. cibaria, were isolated in samples fermented at all temperatures. These species were found only at the early stage of fermentation because their growth was affected by acid [30].

L236-239: sentence not clear, please rephrase

Response: As you recommended, we revised the sentence in lines 260-262 as follows:

Lines 260-262: In the low temperatures fermented samples, Leu. mesenteroides existed until the later stage of fermentation, but in the high temperatures fermented samples, it decreased rapidly as the fermentation process.

L242: L. sakei was increased, please rephrase

Response: As you recommended, we revised the sentence in line 266 as follows:

Line 266: L. sakei increased and predominated the later stage

L251-253: contradictory with the previous sentence. Please rephrase and correct

Response: As you recommended, we deleted the sentence in lines 268-270 as follows:

Lines 268-270: L. plantarum and L. brevis increased in samples fermented at 15°C and 20°C and stabilized in the later fermentation stage, becoming the only dominant species. At low temperatures (5°C and 10°C) this species was not identified.

L265-268: "L. sakei 266 adapted well at low temperatures [28], unlike L. sakei," contradictory with the previous statement. Please rephrase and correct

Response: As you recommended, we revised the sentence in lines 288-290 as follows:

Lines 288-290: L. sakei adapted well at low temperatures [18]. In contrast, L. plantarum (Pearson coeffect r = 0.614, p = 5.914333e-05) and L. brevis (Pearson coeffect r = 0.523, p = 8.662500e-04) demonstrated a positive correlation with fermentation temperature

L269: "these strains", "these LAB species"

Response: As you recommended, we changed “these strains” to “these LAB species” in line 290 as follows:

Line 290: these LAB species

L271-272: sentence not clear, which species?

Response: As you recommended, we revised the sentence in lines 292-293 as follows:

Lines 292-293: L. plantarum and L. brevis are well adapted to acidic environments and high temperatures

L279: "weaker", "weak"

Response: As you recommended, we changed “weaker” to “weak” in line 300 as follows:

Line 300: weak acidic conditions

L289-290: not clear, which bacterium? the previous sentence mention Weisella and Leuconostoc species

Response: As you recommended, we revised the sentence in lines 311-312 as follows:

Lines 311-312: In addition, Leuconostoc and Weissella species were beneficial bacterium that provided the flavor of the fermented foods

L295: use the past tense, cite a reference showing this. Be more specific, what is a low and high temperature?

Response: Low and high temperatures mean fermentation at 5°C/10°C and 15°C/20°C, respectively. As you recommended, we revised the sentence in lines 315-317 as follows:

Lines 315-317: In addition, L. sakei, Leu. gelidum and W. koreensis grew well at low temperatures (5°C and 10°C), whereas L. plantarum and L. brevis grow well at high temperatures (15°C and 20°C).

L299: sentence not clear, L. plantarum is a facultative heterofermentative while L. brevis is an obligate heterofermentative species

Response: Although L. plantarum is a facultative heterofermentative bacterium, previous studies have shown that it grows better under anaerobic conditions than under aerobic conditions. As you recommended, we revised the sentence in lines 320-323 as follows:

Lines 320-323: Furthermore, heterofermentative LAB, such as L. plantarum (facultative heterofermentative species) and L. brevis (obligate heterofermentative species) become dominant as food fermentation conditions change to more anaerobic and acidic conditions [21,31,35].

L304: please rephrase, 20% isolates could not be assigned

Response: As you recommended, we revised the sentence in lines 335-336 as follows:

Lines 335-336: Our studies accurately identified the LAB in gajami-sikhae at the species or genus level using MALDI-TOF MS

L305-306: not clear

Response: We deleted ‘we included only living cells to’ and revised the sentence in lines 335-337as follows:

Lines 335-337: Our studies accurately identified the LAB in gajami-sikhae at the species or genus level using MALDI-TOF MS and observed dominant species.

L310-311: what do you mean by "standardization" ? Be more specific please.

Response: In this study, standardization refers to fermentation conditions (fermentation temperature and period) for gajami-sikhae production. As you recommended, we revised the sentence in lines 341-342 as follows:

Lines 341-342: These results provide information on fermentation conditions (fermentation temperature and period) of gajami-sikhae.

Conclusion: please identify future perspectives of the present study including sensorial analysis, volatile profile analysis, strain selection etc...

Response: As you recommended, we added the sentence in lines 342-345 as follows:

Lines 342-345: The information provided in this study will be useful in developing effective strategies for selecting bacterial strains. Future researches should focus on sensory analysis and volatility profile analysis to improve the quality of gajami-sikhae.

Round 2

Reviewer 2 Report

The authors adequately addressed the reviewers' comments.

Concerning English language and style, I still feel that a revision by a scientific professional English translator is required to improve the overall quality of the manuscript.

Author Response

The authors adequately addressed the reviewers' comments.

Concerning English language and style, I still feel that a revision by a scientific professional English translator is required to improve the overall quality of the manuscript.

Response: As you recommended, we edited the manuscript by MDPI's English editing service (English editing ID: English-41863, MDPI manuscript ID: foods-1626144).